# PETTINGZOO: GYM FOR MULTI-AGENT REINFORCEMENT LEARNING

## ABSTRACT

This paper introduces PettingZoo, a library of diverse sets of multi-agent environments under a single elegant Python API. PettingZoo was developed with the goal of accelerating research in multi-agent reinforcement learning, by creating a set of benchmark environments easily accessible to all researchers and a standardized API for the field. This goal is inspired by what OpenAI's Gym library did for accelerating research in single-agent reinforcement learning, and PettingZoo draws heavily from Gym in terms of API and user experience. PettingZoo is unique from other multi-agent environment libraries in that it's API is based on the model of Agent Environment Cycle ("AEC") games, which allows for the sensible representation of all varieties of games under one API for the first time. While retaining a very simple and Gym-like API, PettingZoo still allows access to low-level environment properties required by non-traditional learning methods.

## 1 INTRODUCTION

Reinforcement Learning ("RL") considers learning a policy — a function that takes in an observation from an environment and emits an action — that achieves the maximum expected discounted reward when acting in an environment, and it's capabilities have been one of the great success of modern machine learning. Multi-Agent Reinforcement Learning (MARL) in particular has been behind many of the most publicized achievements of modern machine learning — AlphaGo Zero (Silver et al., 2017), OpenAI Five (OpenAI, 2018), AlphaStar (Vinyals et al., 2019) — and has seen a boom in recent years. However, popular benchmark environments are scattered across many different locations (or made from scratch), are based around heterogeneous APIs, and are often in unmaintained states. Because of this, highly influential research in the field is generally restricted to institutions with dedicated engineering teams, research into new methods generally aren't compared in like environments, and progress has been slow compared to single agent reinforcement learning (though this obviously cannot be attributed to benchmarks alone).

Motivated by this, we introduce PettingZoo — a Python library collecting maintained versions of all popular MARL environments under a single simple Python API similar to that of OpenAI's Gym library. It's available on PyPI and can be installed via `pip install pettingzoo`.

## 2 A TALE OF TOO MANY LIBRARIES

OpenAI Gym (Brockman et al., 2016) was introduced shortly after the potential of reinforcement learning became widely known with Mnih et al. (2015). At the time, doing basic research in reinforcement learning was a large engineering challenge. The most popular set of environments were Atari games as part of the Arcade Learning Environment ("ALE") (Bellemare et al., 2013). The ALE originally was challenging to compile and install, and had an involved C API and later an unofficial fork with a Python wrapper (Goodrich, 2015). A scattering of other environments existed as independent projects, in various languages, all with unique APIs. This level of heterogeneity meant that reinforcement learning code had to be adapted to every environment (including bridging programming languages). Accordingly, standardized reinforcement learning implementations weren't possible, comparisons against a wide variety of environments were very difficult, and doing simple research in reinforcement learning was generally restricted to organizations with software engineering divisions. Gym was created to promote research in reinforcement learning by making comprehensive

benchmarking more accessible, by allowing algorithm reuse, and by letting average machine learning researchers access the environments. This last point was achieved by putting every environment that a researcher would want to benchmark with (at the time of creation) under one simple API that anyone could understand, in Python (which was just starting to be the *lingua-de-franca* for machine learning). This lead to a mass proliferation of reinforcement learning research (especially at smaller institutions), many environments compliant with the API (Kidziński et al., 2018; Leurent, 2018; Zamora et al., 2016), and many RL libraries based around the API (Hill et al., 2018; Liang et al., 2018; Kuhnle et al., 2017).

In the multiagent space, a similar level of fragmentation currently exists. Notable heterogenous sets of environments include OpenAI's Competitive Multi-Agent Environments for competitive robotic control (Bansal et al., 2017), Sequential Social Dilemma Games for games where cooperation is difficult in a game theoretic sense (Leibo et al., 2017), RLCard for various card games Zha et al. (2019), MAgent for huge numbers of agents (Zheng et al., 2017), Multi-Particle Environments ("MPE") for diverse agent roles (Mordatch and Abbeel, 2017; Lowe et al., 2017), the Starcraft Multi-Agent Challenge (Samvelyan et al., 2019), and dozens more.

## 3 RELATED WORKS

Two attempts at some level of unification in the multi-agent space have been made. The first is Open-Spiel, released by Deepmind in 2019 (Lanctot et al., 2019), which includes excellent implementations of 45 classic games under one sensible API. However, their framework is limited to supporting simple discrete games due to its modeling of games as trees (which is impractical to represent for more continuous environments such as Atari). However, in the space of discrete games it has managed to encourage high quality and fair evaluations of general game solving methods.

The second is the multi-agent API of RLlib Liang et al. (2018), an ambitious distributed RL framework. While the API is powerful, it has very minimal feature support and cannot sensibly represent strictly turn based games (i.e. Hananbi or Go). However, we do maintain PettingZoo support within RLlib so that users can easily leverage the learning methods included on our environments.

## 4 DESIGN PHILOSOPHY

**Simplicity and Similarity to Gym**

The ability for the Gym API to be near instantly understood has been a large driving factor in it's widespread adoption. While a multi-agent API will inherently add complexity, we wanted to create a similarly simple API, and one that would be instantly familiar to researchers who have worked with Gym.

**Agent Environment Cycle Games Based API**

Most environments have APIs that model agents as all stepping at once (Lowe et al., 2017; Zheng et al., 2017; Gupta et al., 2017; Liu et al., 2019; Liang et al., 2018), based on the Partially Observable Stochastic Games (POSGs) model. It turns out this easily results in bugs (Terry et al., 2020b) and is undesirable for handling strictly turn-based games, like chess, since agents aren't allowed to step simultaneously. We instead model our API after the new Agent Environment Cycle games model, an equivalent model where agents step sequentially. That is, an agent performs an action, the environment responds, the next agent acts, the environment responds again, and the cycle repeats. This model allows for the sensible interactions with both strictly turn based games like chess and games where agents truly step simultaneously. A POSG can be easily converted to an equivalent sequential game by having each agent take a step in a cycle, and then updating the rewards and observations of the AEC game at the end of a cycle. For a formal proof of equivalence see (Terry et al., 2020b).

**Variable Number of Agents**

We designed our API to robustly support the widest range of multi-agent scenarios possible, including agent generation and death. No general API currently supports this, but it is such an integral feature of so many environments that this support is essential.

**Sufficient Configurability**

We wanted to make environments that are highly configurable by arguments the norm. In Gym, environments are generally not configurable, and arguments at generation are not used at all. However, playing with various environment properties is often highly desirable, and has accordingly been embraced by Gym environments outside the official library, as this makes research easier and aids reproducibility. Accordingly, we tried to make every reasonable environment parameter an option for users in PettingZoo.

This notion of configuration extends beyond environment configuration to how learning methods interact with the environment. Due to the wide diversity of optimizations and different strategies applied for MARL, we wanted our API to allow for low level access to rewards, observations, done states and other info, while still being very simple for normal applications. Cyclically expansive curriculum learning from Terry et al. (2020b) is a good example of an interesting method that requires this sort of low level access.

**Quality of Life Improvements**

Being users of Gym ourselves, we sought to add several "quality of life" improvements in PettingZoo motivated by frustrations we faced as users. These are:

- Comprehensive, production-grade continuous integration testing. Testing in Gym is arguably lacking compared to other major libraries.

- Tests of environments for API compliance and proper functionality, both for end users and for continuous integration testing of the library. We also provide detailed recommendations for better practices, inspired by the well liked messages of the Rust compiler.

- Good error messages and warnings. When using Gym, triggering an error yields a trace back that needs to be slowly decoded to find the actual problem. We added speciality error messages and warnings for all common errors (that we're aware of) to make development and debugging easier. This is again inspired by the Rust compiler.

- Detailed, comprehensive documentation. Documentation is a fundamental part of a user-friendly software library. Observation space, action space, reward schemes, and other notable environment details are something you generally need to know to begin conducting even the most basic research with an environment. One criticism of Gym is that almost all information is only found in the source code, something especially problematic when working with sets of environments. To solve this in PettingZoo, we created a user friendly wiki-styled website that clearly includes all relevant information for an environment, as well as general information for sets of environments. Our website also includes details about tests, comprehensive API documentation, and so on. This is discussed further in section 7.

## 5 API

### 5.1 MAIN API

Per our discussion above, we sought to create a simple API that could encapsulate all games and be instantly understood to any Gym user, illustrated by comparing Figure 1 and Figure 2. We use the observation/action space objects from Gym, as well as the same seeding method because they are well designed and familiar to researchers.

Figure 1: Basic Usage of Gym

```python
import gym
env = gym.make('CartPole-v0')
observation = env.reset()
for _ in range(1000):
    env.render()
    action = policy(observation)
    observation, reward, done, info = env.step(action)
env.close()
```

Figure 2: Basic Usage of PettingZoo

```python
from pettingzoo.butterfly import pistonball_v0
env = pistonball_v0.env()
env.reset()
for agent in env.agent_iter(1000):
    env.render()
    observation, reward, done, info = env.last()
    action = policy(observation, agent)
    env.step(action)
env.close()
```

## 5.2 PARALLEL API

In addition to our main API, for certain environments we offer a separate API based off of the POSG model. It supports games and methods that assume this model very nicely. In style, it is very similar to RLlib's multi-agent API (Liang et al., 2018), accepting and returning dictionaries keyed on agent names. The primary motivation for including this secondary API is because in games where it's applicable, it can allow for parallelization features which can lead to large performance improvements.

## 5.3 ADDITIONAL API FEATURES

While the API is elegant for simple cases, it's still able to handle important edge cases. This is achieved through lower level API calls for specific attributes: rewards emitted at any time are included for every agent in the `rewards` dictionary attributes, and `dones` and `infos` dictionaries can similarly be used. We additionally have an `observe(agent)` function. `render()` and `close()` function identically to Gym for rendering environments being played by a policy. `agents` is a list of the names of all agents in the game, and due to considerations regarding variable numbers of agents `possible_agents` is an additional list attribute. Finally, `observation_spaces` and `action_spaces` are dictionaries of the Gym observation/action spaces for all agents in `possible_agents`.

When an agent dies, its `done` is set to `True`, it will become the next selected agent to act, and the action taken is required to be `None`. After this dummy step is taken, the agent will be removed from `agents` and all dictionary attributes. Note than a whole environment is done if and only if `agents` is empty. Agent generation can be implemented by simply adding an agent to the agents list and allowing its rewards and observations to be accessed.

Finally, environment configuration is handled by passing arguments to the `.env()` constructor, a standard that all third-party Gym environments have adapted. These arguments can fundamentally change environment behavior, such as the number of agents, the reward structure, and even its action and observation spaces. All environments in PettingZoo allow for some degree of customization in this manner, with many allowing quite a bit.

## 5.4 IMPLEMENTING AN ENVIRONMENT

Compliant environments inherit from a general class (`AECEnv`). To allow for sufficient flexibility, environments only expose lower level attributes (dictionaries of values for all agents — `dones`, `infos`, `rewards`) and an `observe` method that takes an agent. These are then wrapped to provide the more general functions you see above by the base class. This low level functionality allows for entirely new APIs to be efficiently added on top of PettingZoo environments should the need arise. We've done this ourselves with the secondary parallel POSG based API, which have wrappers that convert both to and from the parallel and standard APIs.

## 6 ENVIRONMENTS

Similar to Gym, we wanted to include popular and interesting environments within one package, in an easily usable format. Half of the environment classes we include (MPE, MAgent, and SISL),

Figure 3: Example Environments From Each Class

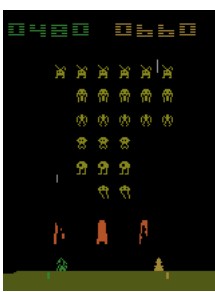

(a) Atari: Space Invaders

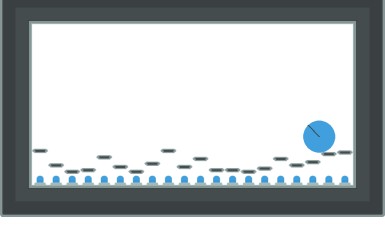

(b) Butterfly: Pistonball

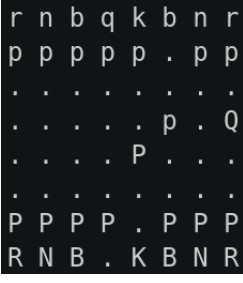

(c) Classic: Chess

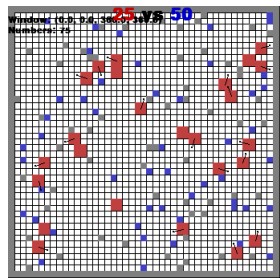

(d) MAgent: Adversarial Pursuit

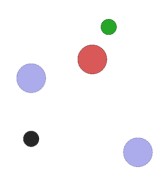

(e) MPE: Simple Adversary

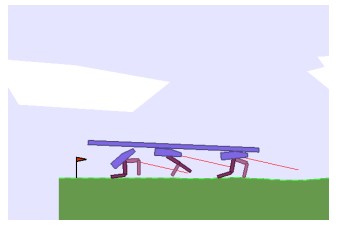

(f) SISL: Multiwalker

despite their popularity, have previously only existed as unmaintained "research grade" code, have not been available for installation via pip, have required large amounts of maintenance to run at all, and have required large amounts of debugging, code review, code cleanup and documentation to bring to a production-grade state. The Multi-Agent Atari and Butterfly classes are new environments that we believe pose important and novel challenges to multi-agent reinforcement learning. Finally, we include the Classic class — classic board and card games popular within the RL literature.

**Atari**

Atari games represent the single most popular and iconic class of benchmarks in reinforcement learning. Recently, a multi-agent fork of the Atari Learning Environment was created that allows programmatic control and reward collection of Atari's iconic multi-player games (Terry and Black, 2020). As in the single player Atari environments, the observation is the rendered frame of the game, which is shared between all agents, so there is no partial observability. Most of these games have competitive or mixed reward structures, making them suitable for general study of adversarial and mixed reinforcement learning. In particular, Terry and Black (2020) categorizes the games into 7 different types: 1v1 tournament games, mixed sum survival games (*Space Invaders*, shown in Figure 3a. is an example of this), competitive racing games, long term strategy games, 2v2 tournament

games, a four-player free-for-all game and a cooperative game. For easy ROM installation, AutoROM, a separate PyPI package, can be used to easily install the needed Atari ROMs in an automated manner.

**Butterfly**

Of all the environments included, the majority of them are competitive. We wanted to supplement this with a set of interesting graphical cooperative environments. *Pistonball*, depicted in Figure 3b, where the pistons need to coordinate to move the ball to the left, while only being able to observe a local part of the screen, requires learning nontrivial emergent behavior and indirect communication to perform well. *Knights Archers Zombies* is a game in which players work together to defeat approaching zombies before they can reach the players. It is designed to be a fast paced graphically interesting combat game with partial observability and heterogeneous agents, where achieving good performance requires extraordinarily high levels of agent coordination. *Cooperative pong*, where two dissimilar paddles work together to keep the ball in play as long as possible, was intended to be a be very simple cooperative continuous control-type task, with heterogeneous agents. *Prison* was designed to be the simplest possible game in MARL, and to be used as a debugging tool. *Prospector* was included to intentionally be a very challenging game for conventional methods—it has two classes of agents, with different goals, action spaces, and observation spaces (something many current cooperative MARL algorithms struggle with), and has very sparse rewards (something all RL algorithms struggle with). It is intended to be an very difficult benchmark for MARL, in the same vein of Montezuma's Revenge.

**Classic**

Classical board and card games have long been some of the most popular environments in reinforcement learning (Tesauro, 1995; Silver et al., 2016; Bard et al., 2019). We include all of the standard multiplayer games in RLCard (Zha et al., 2019): *Dou Dizhu*, *Gin Rummy*, *Leduc Hold'em*, *Limit Texas Hold'em*, *Mahjong*, *No-limit Texas Hold'em*, and *Uno*. We additionally include all AlphaZero games, using the same observation and action spaces—*Chess* and *Go*. We finally included *Backgammon*, *Connect Four*, *Checkers*, *Rock Paper Scissors*, *Rock Paper Scissors Lizard Spock*, and *Tic Tac Toe* to add a diverse set of simple, popular games to allow for more robust benchmarking of RL methods.

**MAgent**

The MAgent library, from Zheng et al. (2017) was introduced as a configurable and scalable environment that could support thousands of interactive agents. These environments have mostly been studied as a setting for emergent behavior (Pokle, 2018), heterogeneous agents (Subramanian et al., 2020), and efficient learning methods with many agents (Chen et al., 2019). We include a number of preset configurations, for example the *Adversarial Pursuit* environment shown in Figure 3d. We make a few changes to the preset configurations used in the original MAgent paper. The global "minimap" observations in the battle environment are turned off by default, requiring implicit communication between the agents for complex emergent behavior to occur. The rewards in *Gather* and *Tiger-Deer* are also slightly changed to prevent emergent behavior from being a direct result of the reward structure.

**MPE**

The Multi-Agent Particle Environments (MPE) were introduced as part of Mordatch and Abbeel (2017) and first released as part of Lowe et al. (2017). These are 9 communication oriented environments where particle agents can (sometimes) move, communicate, see each other, push each other around, and interact with fixed landmarks. Environments are cooperative, competitive, or require team play. They have been popular in research for general MARL methods Lowe et al. (2017), emergent communication (Mordatch and Abbeel, 2017), team play (Palmer, 2020), and much more. As part of their inclusion in PettingZoo, we converted the action spaces to a discrete space which is the Cartesian product of the movement and communication action possibilities. We also added comprehensive documentation, parameterized any local reward shaping (with the default setting being the same as in Lowe et al. (2017)), and made a single render window which captures all the activities of all agents (including communication), making it easier to visualize.

**SISL**

We finally included the three cooperative environments introduced in Gupta et al. (2017): *Pursuit*, *Waterworld*, and *Multiwalker*. *Pursuit* is a standard pursuit-evasion game Vidal et al. (2002) where

pursuers and controlled in a randomly generated map. Pursuer agents are rewarded for capturing randomly generated evaders by surrounding them on all sides. *Waterworld* is a continuous control game where the pursuing agents cooperatively hunt down food targets while trying to avoid poison targets. *Multiwalker* (Figure 3f) is a more challenging continuous control task that is based on Gym's *BipedalWalker* environment. In *Multiwalker*, a package is placed on three independently controlled robot legs. Each robot is given a small positive reward for every unit of forward horizontal movement of the package, while they receive a large penalty for dropping the package.

# 7 DOCUMENTATION

Documentation is a fundamental part of a user-friendly software library. There's a tremendous amount of useful information about these environments, especially due to their diversity, so we sought to create as detailed documentation as possible, while designing it in a way to ensure it's still useful and approachable. PettingZoo includes comprehensive documentation for the API, the continuous integration tests, and each environment. A majority of popular libraries do not have extensive documentation. For example, OpenAI's popular Gym library only lists the observation space shape on each environment's documentation page. PettingZoo's documentation thoroughly explains each environment's observation and action spaces, and includes relevant information to help researchers. The goal is to allow people to compare environments easily, and for developers to very rarely need to refer to source.

Our design for displaying so much information was inspired by Wikipedia's familiar and well-known layout. This is illustrated in Figure 4. All documentation is included in the supplemental materials to facilitate anonymous review.

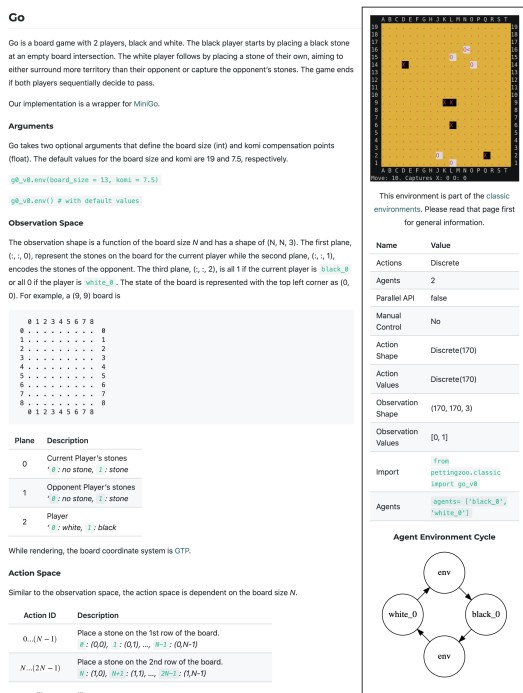

Figure 4: The beginning PettingZoo documentation for the Go environment, illustrating how we used the design metaphor of a Wikipedia page to include a large amount of detail in a manner that isn't overwhelming

# 8 BASELINES

All environments implemented in PettingZoo include baselines to provide a general sense of the difficulty of the environment, and for something to initially compare against. We do this here for the

Butterfly environments that this library introduces for the first time; similar baselines exist in the papers introducing all other environments. We used parameter sharing (Terry et al., 2020c; Gupta et al., 2017) with Ape-X DQN (Horgan et al., 2018), with RLLib (Liang et al., 2018). Our results are shown in Figure 5. Preprocessing and hyperparameter details are included in Appendix A. All preprocessing was done with the SuperSuit wrapper library (Terry et al., 2020a), which has recently added support for PettingZoo based multi-agent environments based. Code for the environments, training logs, and saved policies are available at `https://github.com/pettingzoopaper/pettingzoopaper`.

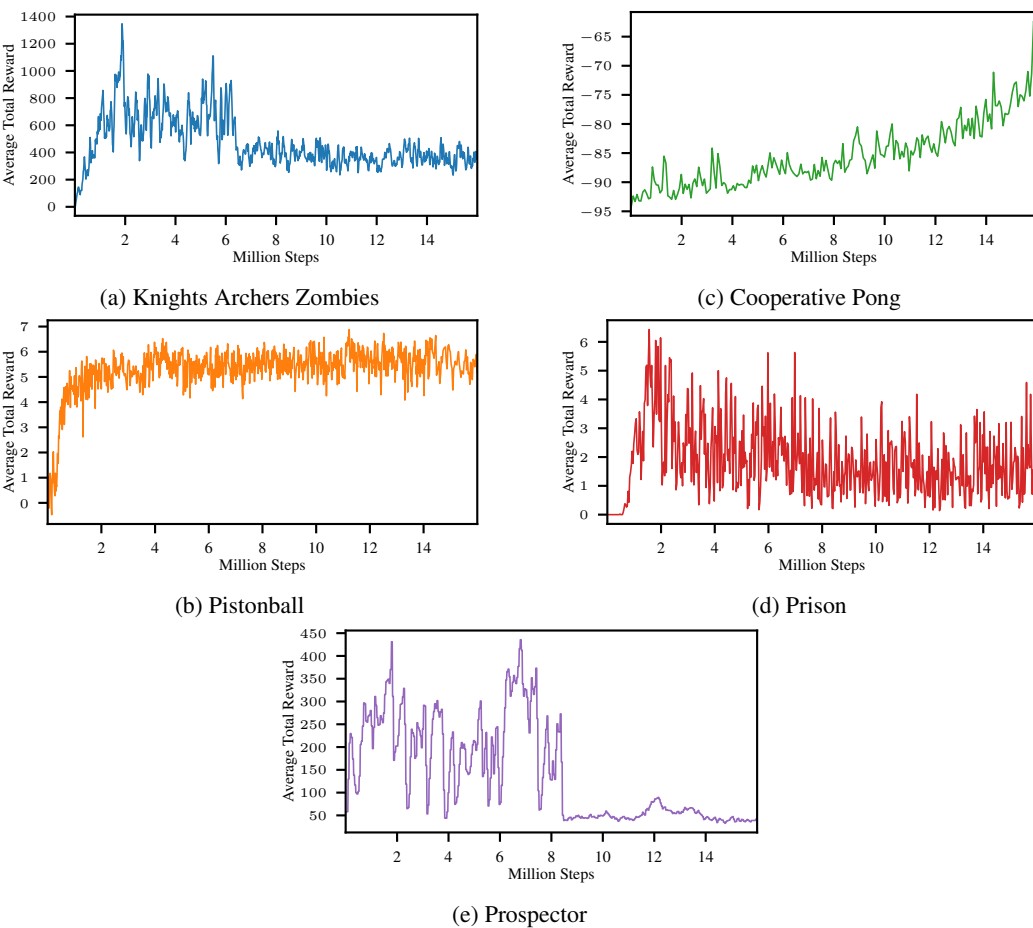

Figure 5: Total reward when learning on each Butterfly environment via parameter shared Ape-X DQN (a-d) and parameter shared PPO (e).

## 9    CONCLUSION

This paper introduces PettingZoo, a Python library of many diverse multi-agent reinforcement learning environments under one simple API, akin to a multi-agent version of OpenAI's Gym library.

Reinforcement learning systems have two main components, the environment and the agent(s) that learn. Without a standardized environment base, research progresses by designing and building both the environment and the agent (as has been the case for MARL). The main contribution of PettingZoo is that it enables more research which focuses on agents by standardizing and democratizing the environments. We hope that this allows for research in multi-agent reinforcement learning to accelerate and flourish.

We're aware of two notable limitations of PettingZoo. The first is that games with significantly more than 10,000 agents (or potential agents) will have meaningful performance issues. This arises

from needing to prespecify observation/action spaces and potential agent names. We view this as a practically acceptable limitation. The second notable limitation is that PettingZoo does not currently allow users to access the global environment state, a feature required by some centralized critic methods. We're actively working on supporting this via a `.state()` method.

We see three obvious directions for future work. The first is additions of more interesting environments under our API (possibly from by the community, as has happened with Gym). While we've included a large number of environments, there are additional sets that would be valuable to include: the open-source implementations of social sequential dilemma games (Vinitsky et al., 2019), and the StarCraft 2 Multi-Agent Challenge ("SMAC") environments (Samvelyan et al., 2019). The second direction we envision is a service to allow different researchers' agents to play against each other in competitive games, leveraging the standardized API and environment set. Finally, we envision the development of procedurally generated multi-agent environments to test how well methods generalize, akin to the Gym procgen environments. (Cobbe et al., 2019).

ACKNOWLEDGMENTS

Thank you to Deepthi Raghunandan and Kevin Hogan for many helpful discussions surrounding what testing should look like. Thank you to Nathaniel Grammel for many helpful discussions in the early planning stages of the project. Thank you to Ross Allen and his group for reporting numerous bugs.

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

## A  BASELINE EXPERIMENT HYPERPARAMETERS AND PREPROCESSING

All of the environments were preprocessed in the following way: observations were resized to 84x84 images with linear interpolation, converted to grayscale, then normalized. This preprocessing was performed with SuperSuit (Terry et al., 2020a).

The graphically subtle environments (Knights Archers Zombies, Prospector and Cooperative Pong) had their observations processed with the RLlib default network: A convolutional layer with a 8x8 kernel, stride of 4, and 16 filters, followed by a convolutional layer with a 4x4 kernel, stride of 2, and 32 filters, followed by a convolutional layer with n 11x11 kernel, stride of 1, and 256 filters.

The graphically simple environments (Prison, Pistonball) were resized to 32x32 and flattened in addition to the above preprocessing. The observation was processed with a network with two hidden linear layers, 400 and 300 neurons wide, respectively.

| RL method | Hyperparameter | Value |
|---|---|---|
| ApeX-DQN | `adam_epsilon` | 0.00015 |
| | `buffer_size` | 400000 |
| | `double_q` | `True` |
| | `dueling` | `True` |
| | `epsilon_timesteps` | 200000 |
| | `final_epsilon` | 0.01 |
| | `final_prioritized_replay_beta` | 1.0 |
| | `gamma` | 0.99 |
| | `learning_starts` | 10000 |
| | `lr` | 0.0001 |
| | `n_step` | 3 |
| | `num_atoms` | 1 |
| | `num_envs_per_worker` | 4 |
| | `num_gpus` | 1 |
| | `num_workers` | 12 |
| | `prioritized_replay` | `True` |
| | `prioritized_replay_alpha` | 0.5 |
| | `prioritized_replay_beta` | 0.4 |
| | `prioritized_replay_beta_annealing_timesteps` | 2000000 |
| | `rollout_fragment_length` | 32 |
| | `target_network_update_freq` | 10000 |
| | `timesteps_per_iteration` | 15000 |
| | `train_batch_size` | 512 |
| PPO | `gamma` | 0.99 |
| | `num_envs_per_worker` | 4 |
| | `num_gpus` | 1 |
| | `num_workers` | 12 |
| | `compress_observations` | `False` |
| | `lambda` | 0.95 |
| | `kl_coeff` | 0.5 |
| | `clip_rewards` | `True` |
| | `clip_param` | 0.1 |
| | `vf_clip_param` | 10.0 |
| | `entropy_coeff` | 0.01 |
| | `train_batch_size` | 5000 |
| | `sample_batch_size` | 25 |
| | `sgd_minibatch_size` | 256 |
| | `num_sgd_iter` | 100 |
| | `batch_mode` | `truncate_episodes` |
| | `vf_share_layers` | `True` |

Table 1: Hyperparameters for ApeX DQN and PPO on each Butterfly environment.

