# OpenReview forum: "PettingZoo: Gym for Multi-Agent Reinforcement Learning"
_ICLR.cc/2021/Conference — Reject_

### Official Review · AnonReviewer3 · 2020-10-16
**Useful library for interacting with multi-agent environments**

**Rating:** 7
**Confidence:** 3

**Review:**

Thanks for time and effort on writing the library.

The paper introduces an API and library for multi-agent reinforcement learning along with simple installation of a very diverse set of environments along. Each environment has clear documentation of inputs/outputs, etc.

Pros:
- The majority of the paper is clear and easy to understand
- Significant effort on the framework
- Clean API
- Integration of very diverse set of environments
- The documentation of individual environments is great

Cons:
- Abstract is very weak. OpenAI Gym is overemphasized, and PettingZoo is underemphasized. Make sure it highlights why PettingZoo is great
- Baselines are weak. Ape-X is far from a good baseline in my opinion. There are simpler more data efficient agents available and also policy gradient based agents. (I realize that these are not the main points of the paper.)
- The paper states it has been shown that AECs are equivalent to POSGs but there is no reference. I assume this is just executing the individual actions of the agents and then getting the observations of all of them instead of interleaving getting observations. Wrt. implementation, a POSG environment used as AEC's may be in invalid states when not all agents have executed an action?
- Paper could be polished more in general.

---

> ### Author Response · Authors · 2020-11-10
> *** Title**
>
> Hey, thank you for your kind words.
>
> -We just picked Apex-DQN because it's an effective and popular DQN based method, and the butterfly environments are very similar to Atari environments in many ways. The goal was just to give a sensible impression of how hard they are to learn. What did you have in mind?
>
> -You aren't the only one to complain about the overemphasis on Gym, we'll upload a version with that cleaned up in the coming days.
>
> -AEC games were proved equivalent to POSGs in the original AEC games paper we cite, we'll clarify that.
>
> -"Wrt. implementation, a POSG environment used as AEC's may be in invalid states when not all agents have executed an action?" I don't think I quite follow what you mean by that, could you please elaborate?

---

> > ### Comment · AnonReviewer3 · 2020-11-12
> > **Comments**
> >
> > -We just picked Apex-DQN because it's an effective and popular DQN based method, and the butterfly environments are very similar to Atari environments in many ways. The goal was just to give a sensible impression of how hard they are to learn. What did you have in mind?
> >
> > For Q learning, I would have preferred to see something like Rainbow (non-distributed). Could also have been something like PPO if considering policy gradient methods.
> >
> > -"Wrt. implementation, a POSG environment used as AEC's may be in invalid states when not all agents have executed an action?" I don't think I quite follow what you mean by that, could you please elaborate?
> >
> > Please ignore.

---

> > > ### Author Response · Authors · 2020-11-13
> > > *** Title**
> > >
> > > Thanks for clarifying there.
> > >
> > > We used Apex-DQN with all the rainbow optimizations turned on. The parallelization features Apex-DQN ads just save on AWS costs is all, they're available in a few major RL library. We did use PPO for Prospector though, since it's a parallel only environment.
> > >
> > > We also uploaded an updated version of the paper with those two fixes.

---

> > > > ### Comment · AnonReviewer3 · 2020-11-13
> > > > **Sample efficiency of baseline**
> > > >
> > > > Thanks, will take a look at the updated paper.
> > > >
> > > > If you used Ape-X in a distributed fashion like in the original paper, then you will experience a much worse sample efficiency as can be seen in Appendix on page 17 in https://openreview.net/pdf?id=H1Dy---0Z . The sample efficiency will depend on the number of machines used and is quite a bit worse than the Rainbow baseline. That, in my view, makes Ape-X a poor baseline to use.

---

### Official Review · AnonReviewer4 · 2020-10-21
**Overall an ok paper with an important contribution to the ecosystem of MARL frameworks, let down by focus on the wrong areas.**

**Rating:** 5
**Confidence:** 4

**Review:**

### Quality
The paper is overall a good quality but it has some deficiencies. The authors need to do a better job comparing PettingZoo to other alternatives. What's the selling point?

### Clarity
The paper was straight forward and easy to read.

### Originality
While obviously taking inspiration from OpenAI Gym, others have also thought about MARL. However the drivers which led to the development of PettingZoo are original with a focus on quality of life improvements and configurability. These are all positives.

### Significance
There is an obvious interest in MARL approaches and hence PettingZoo can be a potentially significant contribution

## Comments on specific sections

#### Abstract:
80% of the abstract is talking about OpenAI Gym, not about PettingZoo. The last sentence then mentions PettingZoo almost in passing. I recommend you start with Petting Zoo and say that it was inspired by OpenAI Gym and focus your abstract on what your paper is about - Petting Zoo.

#### Introduction
I have a similar issue with the introduction. You spend a lot of time extolling the virtues of OpenAI Gym, which is great because I can see that Petting Zoo has taken inspiration from Gym, however this could be put into a background or related work section.
Your introduction should focus on introducing PettingZoo, describing the scope of the paper, an overview of the contributions made and giving the reader an overview of what is coming in the rest of the paper.

#### Background
As mentioned above, the paper is missing a dedicated background or related work. You mention RLlib as an alternative multi-agent RL library but you don't go into detail. How about other MARL libraries/frameworks? What is the difference between PettingZoo and these? Advantages/Disadvantages of other systems? Why not contribute to OpenAI Gym?


#### Design Philosophy
I liked the design philosophy described. This list can also be thought of as motivation or key development drivers. This potentially could be your development methodology. However what is missing is evaluation criteria. Once you've developed PettingZoo what are the measurable qualitative or quantitative criteria that you will/can use to evaluate if you have done a good job.

#### API
I feel you should have gone into more detail into your MARL API. I had a look at the code and there is a bit more of the API which would be good to talk about especially the parallel API. A design or block diagram showing the architecture of

#### Environments
Its good to see a comprehensive set of environments available with the PettingZoo library.

#### Documentation:
While this is good you probably don't need to spend 3/4 of a page on it.

#### Baselines:
The baselines are good, but it would be nice to compare them to other MARL libraries as well.

### Other Comments
The paper is missing a discussion/evaluation section where the authors critically evaluate the PettingZoo library.With some of the changes suggested the paper will be much stronger.

---

> ### Author Response · Authors · 2020-11-13
> *** Title**
>
> I really appreciate all your detailed feedback. Genuinely.
>
> I just uploaded a revised version of the paper that I believe addresses all of your points but two:
>
> -The big documentation figure is still in there because we have the space, and it's a really cool thing and a key selling point of PettingZoo. Also no one reads appendixes.
>
> -"The baselines are good, but it would be nice to compare them to other MARL libraries as well." No other libraries than RLlib have serviceable multi-agent learning support. We're actually currently adding it to a major single agent RL library for that reason, but that's not part of this paper.
>
> An additional note: I took a slightly different approach to a proper evaluation section. If I did it right, the paper should now clearly address how each component of the design philosophy was individually satisfied through the paper. I did this because, while this sort of thing is common in proper engineering papers, I can't find any library paper in a proper ML venue does this and the same information is included either way.

---

### Official Review · AnonReviewer1 · 2020-10-25
**Interesting contribution, could use some more detail**

**Rating:** 6
**Confidence:** 3

**Review:**

Summary
---

The paper describes a new framework, "PettingZoo", which is proposed to play a similar role for Multi-Agent RL research as OpenAI's Gym framework does for single-agent RL research. The paper describes various lessons learned from Gym and other frameworks, and how these were taken into account in PettingZoo's design and implementation.

Strong Points
---

1) These types of frameworks for standardised, easy-to-use, and varied benchmarks are important contributions.
2) Framework looks easy to use, familiar API for Gym users.
3) Paper well-written, easy to follow.
4) Nice detailed listing of all hyperparameters in Appendix for the included baseline results.

Weak Points
---

The paper could use a more thorough review of existing work and a more detailed comparison to make a stronger case for this framework satisfying a need in the research community that is not already satisfied. The primary point seems to be the different "Agent Environment Cycle" model from (Terry et al., 2020b). Since this is from a very new paper (this year), not widely-known already (to the best of my knowledge), and seemingly a primary core point of this paper's contribution, it would probably be useful to elaborate on exactly what this model is and how it's different from others a bit more. There is a very short description, but from that one I can't see how it's any different from Extensive Normal Form games (with imperfect information). If it is indeed very similar / identical to the standard Extensive Normal Form games model, it would probably also be worthwhile to discuss other frameworks that already use these kinds of models in related work. Especially OpenSpiel comes to mind, since it has a wide variety of Multi-Agent RL algorithms built-in (and many different kinds of games), but possibly even other general game playing systems may be worth comparing to (such as Polygames, Ludii, GDL, etc.). Many of these have different focuses, but may still be worth comparing to them.

Overall Recommendation
---

Currently I'm leaning towards marginally above the acceptance threshold, because I do see potential and feel like the contribution is interesting and there's nothing really "wrong" with the paper. More details (see above) could definitely be useful to add though.

Questions for Authors
---

Could you elaborate on how this Agent Environment Cycle games model is different from Extensive Normal Form games? I understand that this is not the contribution of this paper per se, but it does seem to be a core motivation point for why this new framework is better than existing ones?

Minor Comments
---

- In abstract: "wrapper" --> "wrappers for" or "wrapping" (I guess)
- Variable names should be kept consistent between python code in Figures 1 and 2. Now Figure 1 has "obs" but Figure 2 has "observation". Figure 2 also explicitly assigns the name "agent" to the variable it loops over, but this variable isn't used, so can just stick to "_" like in Figure 1 I suppose?
- Section 4 starts naming a bunch of acronyms (MPE, MAgent, SISL), but I don't really know what any of those mean.
- Inconsistent formatting for the "Classic" paragraph in Section 4 (text starts immediately after paragraph header instead of below it)

---

> ### Author Response · Authors · 2020-11-13
> *** Title**
>
> Hey, thank you for your feedback.
>
> The primary difference of PettingZoo and other libraries is its ability to cleanly include all MARL environments, and doing so in a "production grade" state. Modeling the games through the AEC games paradigm allows this to be possible. AEC games are proven in the paper introducing them to be equivalent to partially observable stochastic games, however the mechanics of them differ. AEC games are ultimately just partially observable stochastic games. The relationship between POSGs and AEC games is very similar to EFGs/NFGs in that they're equivalent formulations that are useful in different scenarios. The difference between POSGs and EFGs, aside from the mechanical description, is just how often rewards are usually thought of as being emitted.
>
> You're absolutely right though that the AEC games section needed a lot more detail; we wrote it from the perspective of people who were extraordinarily familiar with it.
>
> We just uploaded a new revision of the paper. It explains AEC games in adequate detail for someone unfamiliar with the model,  adds a more clear related works that we discuss OpenSpiel in (and explain why it can't do what PettingZoo does on a technical level), and should address all the other concerns you raised.

---

### Official Review · AnonReviewer2 · 2020-10-27
**Yet another testing framework for RL**

**Rating:** 3
**Confidence:** 3

**Review:**

The paper introduces yet another testing environment for RL, PettingZoo, as a logic evolution of OpenAI but, this time, for multi-agent RL systems, a feature not supported by the former.

Reasons to Accept:
•	Very nice engineering effort. The platform includes popular environments in a user-friendly way as well as detailed documentation and baselines for comparison.
Reasons to reject:
•	Contents: I am confused by the nature of the work and assume it is meant as a demo. Actually, the paper reads like a product/company brochure, not as a scientific paper. This is mostly due PettingZoo is a great engineering effort (involving the use/incorporation of SW/packages from outside) with little science behind. Sections 2 to 4 gives an abstract overview of the main mechanisms in the API, environments and the interaction with the user, but all basically at the level of interaction at code level. The paper itself is not interesting from a research point of view, for me, because it does not provide any AI-like content apart from summarising the environments/APIs. I think sections 2-4 could have been compressed to one section introducing PettingZoo, after which an actual research contribution using the system could follow, to create a proper research paper. Further explanations, discussions and insights are thus needed (e.g., further comparisons with SOTA MARL approaches/models for different environments).
•	Relevance/Novelty: I also wonder whether the paper really represents a significant advance in the AI field, and I guess its relevance may be below the ECLR threshold. Even though I personally don’t see any particularly novel insight in this paper (it is a logic incremental evolution of OpenAI's Gym), the software coding/integration methodology followed is not wrong and the whole environment seem promising and may lead a mass proliferation of MARL research. And I happy to let the noisy process of science (and reviewing process) figure out the value here. I am fully open to change the score if the authors and the rest of reviewers convince me of the usefulness and relevancy of the contribution.
•	Suitability: Finally, I like the whole system (I actually think it is great and has great potential to be used as testbed for MARL system), but I feel this could have been better put forward at a dedicated workshop, as an overview, as an unpublished introduction paper, or as a white paper or technical report to briefly describe a system.

---

> ### Author Response · Authors · 2020-11-10
> *** Title**
>
> You're absolutely correct that this is an engineering paper. However, this does not make it inherently unfit for ICLR. ICLR's call for papers requests papers concerning "implementation issues, parallelization, software platforms, [and] hardware" for machine learning, and PettingZoo is a software platform for machine learning. Many other large software platforms have been published as full papers in machine learning conferences before too (e.g. PyTorch at NeurIPS), and they all look pretty similar to our paper.
>
> Regarding community importance, what I can say here without wild violations of anonymity is limited. However, PettingZoo was publicly released in August. Since then numerous research groups from at least 3 continents have replaced all their internal tooling with PettingZoo. PettingZoo has also gone viral on an ML focused social media community, has been accepted to a major RL workshop, and we've received about 6 pieces of proper fan mail on it. It's also worth noting that no other reviewers found this aspect concerning, they only had concerns regarding the writing of the paper.
>
> Also, I really like your recommendation on running a dedicated workshop. I'm going to seriously look into that.

---

### Decision · Program_Chairs · 2021-01-07
**Final Decision**

**Decision:**

Reject

**Comment:**

This paper represents the PettingZoo library of multi-agent environments, providing a common API and benchmark for multi-agent learning. The library has high potential for impact and is likely of interest to a wide range of people in the ICLR community. However, in its current form the paper could be significantly improved by actioning the many pieces of constructive feedback provided by all reviewers.

We have also been made aware of two highly related papers "Multiplayer Support for the Arcade Learning Environment" and "SuperSuit: Simple Microwrappers for Reinforcement Learning Environments." Together all three papers could be one comprehensive manuscript, but appear to have been unnecessarily split into three separate short papers.

---

> ### Author Response · Authors · 2021-01-15
> **Note regarding paper splitting**
>
> The AC's comment regarding unnecessarily splitting the paper is trivially wrong if you actually skim the mentioned papers.
>
> The AC actually errored so gravely in their assessment of our similarity to other works that the PCs took the extraordinary action of forcing them to retract the majority of their criticism, as it was public. The only reason the this single comment remains is the PCs determined it was a "technical criticism", and that it was therefore against conference policy to amend.